# Synthesis and Catalytic Performance of High-Entropy Rare-Earth Perovskite Nanofibers: (Y_0.2_La_0.2_Nd_0.2_Gd_0.2_Sm_0.2_)CoO_3_ in Low-Temperature Carbon Monoxide Oxidation

**DOI:** 10.3390/ma17081883

**Published:** 2024-04-19

**Authors:** Paweł A. Krawczyk, Jan Wyrwa, Władysław W. Kubiak

**Affiliations:** Faculty of Materials Science and Ceramics, AGH University of Krakow, Al. Mickiewicza 30, 30-059 Kraków, Poland; wyrwa@agh.edu.pl

**Keywords:** high-entropy perovskites, glycothermal synthesis, carbon monoxide oxidation

## Abstract

This study investigated the catalytic properties of low-temperature oxidation of carbon monoxide, focusing on (Y_0.2_La_0.2_Nd_0.2_Gd_0.2_Sm_0.2_)CoO_3_ synthesized via a glycothermal method using 1,4-butanediol and diethylene glycol at 250 °C. This synthesis route bypasses the energy-intensive sintering process at 1200 °C while maintaining a high-entropy single-phase structure. The synthesized material was characterized structurally and chemically by X-ray diffraction and SEM/EDX analyses. The material was shown to form nanofibers of (Y_0.2_La_0.2_Nd_0.2_Gd_0.2_Sm_0.2_)CoO_3_, thereby increasing the active surface area for catalytic reactions, and crystallize in the model Pbnm space group of distorted perovskite cell. Using a custom setup to investigate catalytic properties of (Y_0.2_La_0.2_Nd_0.2_Gd_0.2_Sm_0.2_)CoO_3_, the CO oxidation behavior of those high-entropy perovskite oxide was investigated, showing an overall conversion of 78% at 50 °C and 97% at 100 °C. These findings highlight the effective catalytic activity of nanofibers of (Y_0.2_La_0.2_Nd_0.2_Gd_0.2_Sm_0.2_)CoO_3_ under mild conditions and their versatility in various catalytic processes of robust CO neutralization. The incorporation of rare-earth elements into a high-entropy structure could impart unique catalytic properties, promoting a synergistic effect that enhances performance.

## 1. Introduction

Continuing technological advances underscore the need to develop materials capable of meeting the growing demands of modern engineering. This covers not only performance and cost considerations, but the environmental impact, which is key to ensuring a sustainable future [1]. In response to this challenge, the exploration of high-entropy oxides with their inherent complexity and ability to tune properties represents a new strategy in advanced materials development [2,3]. Since their initial discovery by Rost et al. in 2015 [4], high-entropy oxides exploration has experienced notable expansion, particularly following the discovery of multicomponent rare-earth oxides (ME-REOx) [5]. This finding expanded the possibilities, allowing for the development of more complex high-entropy structures, including garnets and perovskites [6,7,8]. Known for their versatility in both bulk [9,10] and thin-film configurations [11], perovskite-type high-entropy oxides exhibit distinctive properties that make them highly attractive in a variety of applications covering technological domains from photovoltaics [12,13,14,15] to electronics [16,17,18,19], to a new class of catalytic materials [20,21,22,23]. To improve the properties of the mentioned catalytic materials, downsizing is essential. The strategy of employing novel nanostructuring methods demonstrates its efficiency in enhancing electrocatalytic performance and obtaining high-performance perovskite nanocatalysts [24,25].

The synthesis of multicomponent perovskites with stable entropy is a major challenge due to their complexity compared to simpler oxide structures such as rock salt [26,27]. Conventional synthesis methods, namely solid-state reaction or precipitation methods, often yield heterogeneous or unstable materials [28]. Alternative methods, such as nebulized spray pyrolysis [8] or combustion techniques [29], while offering potential benefits, may be limited by scalability, complexity, or cost [30,31]. Furthermore, the growing demand for efficient catalysts, particularly in the area of low-temperature CO catalysis for applications in power transmission and distribution, underscores the critical need for advances in materials synthesis [32]. The current knowledge gap focuses on identifying high-performance absorbers and catalytic materials, while addressing issues of scalability and complexity are crucial [33]. Perovskite-type catalysts, with their ability to stabilize different oxidation states and induce structural distortions, have the potential to meet these challenges in catalytic technologies [34].

In our previous research, we have developed an efficient method to synthesize high-entropy perovskites using hydrothermal synthesis with quenching in liquid nitrogen [35]. However, a significant drawback of this approach is its high energy consumption, which limits its practicality for large-scale industrial applications. Recognizing the need for more energy-efficient synthesis methods that could potentially be implemented in future industries, we investigated several approaches to minimize energy consumption in the synthesis of perovskites. One of the approaches we explored involved the use of a glycothermal method, which offers the advantage of synthesizing nanomaterials at a relatively low temperature of 250 °C [36]. Unlike the hydrothermal method, the glycothermal approach eliminates the need for high-temperature calcination steps after post-treatment to remove organic residues and form a stable single-phase material from a precursor [37]. By exploiting the unique capabilities of glycothermal synthesis, we sought to contribute to the development of more sustainable and energy-efficient material synthesis techniques for future industrial applications.

In this context, our study aims to investigate the efficiency, performance and reproducibility of glycothermal synthesis methods for perovskite oxides of rare-earth and transition metals, in particular (Y_0.2_La_0.2_Nd_0.2_Gd_0.2_Sm_0.2_)CoO_3_, with a focus on their catalytic properties in CO oxidation. Building on previous studies, here we present an approach using glycol dispersants to investigate the enhancement of catalytic properties of this material in CO oxidation reactions by nanostructuring. The main results of this study underscore the effectiveness of (Y_0.2_La_0.2_Nd_0.2_Gd_0.2_Sm_0.2_)CoO_3_ nanofibers in catalyzing CO oxidation under mild conditions and highlight the potential for future studies of incorporating rare earth elements into high-entropy structures to further enhance and explore their catalytic performance.

## 2. Materials and Method

### 2.1. Synthesis

Synthesis of rare and transition metal oxides of the perovskite type with high-entropy, in particular RECO (RE=Y, La, Nd, Gd, Sm, C=Co, O=O_3_), was carried out by a glycothermal method. Initially, precursor salts, including (CH_3_COO)_3_Y, (CH_3_COO)_3_La, (CH_3_COO)_3_Nd, (CH_3_COO)_3_Sm, (CH_3_COO)_3_Gd and (CH_3_COO)_3_Co, were mixed with a solvent mixture of 1,4-butanediol (BD) and diethylene glycol (DEG) in a volume ratio of 9:1. The resulting colloidal solution was stirred on a hot plate for 3 h at a mild temperature of 50 °C. The suspension was then further homogenized using an ultra-fast homogenizer (IKA Ultra Turrax) to ensure a smooth and well-homogenized mixture.

The homogenized suspension was then transferred to a 500 mL high-pressure reactor system (Büchi AG, Flawil, Switzerland) connected to dry argon to degas the suspension. Continuous stirring was maintained in the reactor at 500 rpm, heating it to 250 °C for 72 h. The desired reaction temperature was achieved by gradually increasing the temperature at a rate of 1 K min^−1^, resulting in an internal pressure of about 50 bar in the reactor. After the process was completed, the mixture was cooled. To remove organic by-products such as 1,4-dioxane and tetrahydrofuran, as well as dispersants (BD and DEG), the suspension underwent high-speed centrifugation at 10,000 rpm. Subsequently, the resulting precipitate was dried in a vacuum oven at 120 °C, yielding the base material of RECO, hereafter referred to as GT_250°C_.

Although an additional heat treatment step was not necessary to achieve the desired high-entropy perovskite structure, a subset of samples was subjected to an additional firing step at 1200 °C in air for comparative analysis. This process mirrored the methodology used in the hydrothermal quenching study, resulting in samples labeled as RECO GT_1200°C_.

Estimation of energy consumption in the glycothermal synthesis process was carried out using an Inepro PRO1S single-phase energy meter (Nieuw-Vennep, The Netherlands). The measurement was carried out by continuously recording the integrated power recalculated output in kWh. The measurement process included electricity consumption for each step of glycothermal synthesis, including processing of the colloidal precursor solution, glycol reaction in the high-pressure reactor system, centrifugation and drying steps. For comparison purposes, energy consumption measurements were also recorded for the model hydrothermal synthesis of RECO material described in our previous study [35]. In this case, energy consumption measurements were made in addition to the aforementioned steps for the high-temperature treatment step in a furnace (SentroTech, Strongsville, OH, USA) at 1200 °C for 2 h. 

### 2.2. Material Characterization

Powder analysis via X-ray diffraction was conducted at room temperature using a Bruker D8 diffractometer (Billerica, MA, USA) equipped with Bragg-Brentano geometry and Cu-Kα radiation. Data collection spanned a diffraction angle range from 10 to 90°, with a step size of 0.015° and a collection time of 5 s per step. Rietveld refinement was carried out using Fullprof software (version 7.40) [38]. The RECO perovskite phase was refined based on the standard orthorhombic structure of GdFeO_3_-type perovskite (*Pbmn*, ICSD 23823) retrieved from the Open Crystallographic Database [39]. The instrumental intensity distribution for the XRD data was determined using the LaB_6_ reference scan (NIST 660a, Argonne National Laboratory, Lemont, IL, USA).

The microstructure and elemental composition of the synthesized powders were examined using a scanning electron microscope (SEM) (Tescan VEGA 3, Brno, Czech Republic) operating at 20 kV. The SEM was equipped with an energy dispersive X-ray spectroscopy (EDX) system from Oxford Instruments X-Max 50 silicon drift system, coupled with AZtec INCA software (version 3.9). Quantitative EDX data were collected from 20 randomly distributed points for each sample, while elemental mapping was obtained from a square image measuring 10 × 10 µm. The resulting values were averaged and expressed along with the standard deviation. 

For particle size distribution measurement for RECO GT_250°C_, dynamic light scattering (DLS) analysis was conducted using a 12 mm cell (DTS 0012). Initially, one milliliter of ethylene glycol was introduced into the cell, followed by the addition of 50 μL from stock dispersions containing approximately 1 wt-% of the material under investigation. Subsequently, the samples underwent sonication for 5 min to ensure proper dispersion.

Specific surface area of powders was obtained from nitrogen adsorption isotherms at 77K using a semi-automatic device (Horiba SA-9600, Osaka, Japan). Prior to adsorption measurements, the samples were degassed for 16 h at 25 °C, achieving a residual pressure of below 1 mPa. Oxygen-free nitrogen with a purity of 99.999% (PanGas AG, Dagmersellen, Switzerland) served as the adsorbent during the experiments.

### 2.3. CO Oxidation Tests

A Nicolet iS50 FT-IR spectrometer (Thermo Scientific, Waltham, MA, USA) equipped with a liquid nitrogen-cooled MCT detector and operating at a resolution of 4 cm^−1^ was used to measure the CO concentration during the catalytic reaction, and measurements were carried out in transmission mode using a 2000 mm path-length gas cell (PIKE Technologies, Stanmore, UK). The sample, weighing 500 mg, was placed in a stainless-steel autoclave on a meshed stainless-steel base and placed in a small cylindrical holder equipped with a ring oven and a K-type thermocouple to monitor and control the temperature of the oxidation catalysis reaction. The gas inlet to the cell was connected to a gas manifold system with calibrated mass flow controllers (Enfield Technologies, Trumbull, CT, USA) to regulate the gas flow between the autoclave and the gas cell. The autoclave was filled with a gas mixture with an absolute pressure of 2000 mbar, consisting of a calibrated gas mixture from PanGas AG containing 1.00 mol-% CO, 5.02 mol-% O_2_ in N_2_. Infrared spectra were collected in the 4000–900 cm^−1^ range by averaging 256 scans to obtain a favorable signal-to-noise ratio. Background subtraction for all spectra was performed using the spectrum of an empty sample holder recorded in nitrogen. Quantitative measurements of CO and CO_2_ concentrations were conducted using calibration standards supplied by PanGas AG, covering a range from 1.00 to 0.10 mol-% CO in CO_2_ and from 5.01 to 0.10 mol-% of CO_2_ in N_2_. Calibration was performed using the heights of the absorption peaks at 2120 cm^−1^ and 2309 cm^−1^ for CO and CO_2_, respectively. Measurements were taken after heating the reactor to temperatures of 25, 50, and 100 °C, with each temperature maintained for a duration of 50 min.

## 3. Results

The XRD patterns of powders synthesized by glycothermal methods, both with heat treatment at 1200 °C (RECO GT_1200°C_) and without treatment but mechanically purified after low-temperature glycothermal synthesis (RECO GT_250°C_), are shown in Figure 1. The main peaks observed in both samples correspond to an orthorhombic structure with the Pbnm space group perovskite, indicating the successful formation of the desired phase. No significant difference was observed between the XRD patterns of samples synthesized by glycothermal methods with and without subsequent heat treatment at high temperatures. However, upon closer examination of the XRD patterns, subtle differences emerged. It was noted that there were some contributions to the background, probably coming from nanostructured RECo_3_O_9_ phase(s). Notably, slightly broader characteristic peaks were observed in the RECO GT_250°C_ sample at similar diffraction angles compared to the sample treated at 1200 °C. This observation suggests that samples synthesized at lower temperatures may also exhibit preferred crystallite orientation and peak broadening due to nanoscale crystal domains [40]. The occurrence of multiple phases in the high-temperature-treated sample can be attributed to the relatively low cooling rate, estimated at 20 °C min^−1^. Consequently, thermodynamically favored phases, i.e., RECo_3_O_9_, that are stable at room temperature may have been formed by unrestricted crystallization, a phenomenon previously demonstrated in our previous study [35].

Figure 2 shows the Rietveld refinement of the XRD pattern obtained from the RECO GT_250°C_ sample, along with the lattice parameters and structural positions refined from the pattern collected at T = 298 K. The analysis reveals that the sample conforms to the desired Pbnm space group (No. 62) with a phase purity of at least 99%.

Figure 3a shows a scanning electron microscopy (SEM) image showing the morphology of the sample synthesized at 250 °C by glycothermal (RECO GT_250°C_). The image shows that the synthesized material consists of uniform nanofibers of submicron length. Using the data derived from the SEM image, Figure 3b illustrates the calculated average particle length of 0.62 ± 0.16 µm, with an average diameter of 48 ± 5 nm. The specific surface area measured for RECO GT_250°C_ was notably high, at 27.3 m^2^ g^−1^. It is worth noting that after heat treatment, the material underwent a significant morphological transformation, characterized by pronounced consolidation due to melting and recrystallization occurring during the free cooling process, which resulted in the appearance of dendrite-like structures in the sample, as shown in Figure 3c. In addition, the specific surface area significantly decreased to 1.2 m^2^ g^−1^. Analysis of the particle size distribution of the RECO GT_1200°C_ sample showed a further shift toward larger particle sizes. The EDX mapping results, shown in Figure 3e, reveal a uniform distribution of all elements throughout the matrix, indicating the absence of segregation or clustering phenomena. This observation underscores the homogeneity of the sample composition at the microscale. In particular, no clear tendencies toward segregation or formation of secondary phases were observed.

To further confirm the homogeneity of the perovskite sample, EDX point elemental analysis was performed on the selected crystallites and grains seen in Figure 3a,c (points 1 and 2) and (points 3 and 4), and the results are summarized in Table 1. In addition, EDX analysis was performed for 20 random points on each sample, and the average composition obtained from these analyses is shown in Table 1, in rows of average atomic concentration. The elemental composition obtained is consistent with the expected chemical composition for both the RECO GT_250°C_ and GT_1200°C_ samples. Any deviations from the nominal formula are within the accuracy of the EDX technique, typically within a few percent, given the complexity of the chemical composition analyzed.

To assess the catalytic activity of the synthesized fibrous materials, a custom-designed method was developed to evaluate the oxidation performance of carbon monoxide (CO) to carbon dioxide (CO_2_), as described in paragraph 2. Figure 4a–c show the evolution of Fourier transform infrared (FTIR) spectra collected at temperatures from 25 °C to 100 °C over a 50 min period. The obtained spectra provide a visual representation of the progressive formation of CO_2_, as evidenced by the appearance and intensification of absorption bands associated with asymmetric C-O stretching in the range 2300–2400 cm^−1^ over time, reflecting the oxidation of CO to CO_2_. In addition, the presence of bands assigned to the stretching modes ν_1_ (symmetric stretching) in the range of 2040–2280 cm^−1^ and ν_2_ (asymmetric stretching) in the range of 2150–2200 cm^−1^, characteristic of the CO molecule, indicates a decrease in its concentration over time at each temperature, with a simultaneous increase in CO_2_ concentration. This observation confirms the progress of CO oxidation to CO_2_ in a closed experimental system. The CO_2_ output concentrations resulting from the oxidation of 1.00% molar CO in N_2_ provide the possibility to evaluate the catalytic reaction yield of RECO GT fibers at 250 °C during CO oxidation. At an ambient temperature, the reaction efficiency is relatively low, with a reaction efficiency of 18% observed after 50 min. However, a significant increase in CO oxidation efficiency is seen at elevated temperatures of 50 °C and 100 °C. In particular, a significant reduction in CO concentration was observed from an initial 1.00 mol-% to 0.22 mol-% and 0.03 mol-% after 50 min, highlighting the temperature-dependent catalytic performance of the RECO fibers.

## 4. Discussion

Although the glycothermal synthesis method used here was not originally designed for this purpose, our results suggest its suitability and cost-effectiveness for the production of nanofibers. In particular, the use of 1,4-butanediol and diethylene glycol in this process, carried out at a relatively low temperature of 250 °C, has proven effective, offering promising results for the synthesis of such materials. Significant morphological transformations were observed during the heat treatments, conducted at temperatures typical for the fabrication of high-entropy oxide materials at 1200 °C. In particular, the material underwent significant densification and recrystallization. These changes underscore the advantages of low-temperature synthesis methods, not only in terms of energy savings, but also in facilitating the manufacturing of nano- or submicron-scale materials with a large specific surface area, which is a critical factor in the fabrication of materials for efficient catalysis. This versatility of synthesis parameters opens many possibilities for diverse applications and precise control of material properties.

The structural and compositional analyses carried out in this study, including XRD diffraction, Rietveld refinement and EDX point analysis and mapping, collectively confirm our findings, providing evidence of the phase purity of (Y_0.2_La_0.2_Nd_0.2_Gd_0.2_Sm_0.2_)CoO_3_ synthesized by the glycothermal method at 250 °C. The energy consumption of the glycothermal process was estimated to be about 52% lower than the hydrothermal synthesis method used in our previous study [35]. This significant reduction in energy consumption underscores the potential energy-saving benefits offered by the glycothermal approach, in line with the emphasis on energy efficiency emphasized throughout this research. It is noteworthy that our results show an even higher degree of phase purity compared to the same material synthesized in our previous study using a hydrothermal co-precipitation method, in which about 4 wt-% of the precipitated RE_2_O_3_ phase was identified [35]. Conventionally, current high-entropy oxides are typically limited to high-density materials with minimal defects, requiring high synthesis temperatures ranging from 1000 to 1200 °C to accentuate the entropic contribution (TΔS) in the Gibbs free energy equation: ΔG = ΔH − TΔS, (1)
where: ΔG—Gibbs free energy, ΔH—enthalpy, T—temperature in Kelvin, and ΔS—entropy. However, as demonstrated by an approach showing that the negative Gibbs free energy for high-entropy perovskite, crystallization can be achieved by significantly reducing the enthalpy of mixing (ΔH) [42]. This novel approach is in line with recent research [43], indicating that high-entropy oxides can be synthesized at lower temperatures, as exemplified by successful synthesis of (Y_0.2_La_0.2_Nd_0.2_Gd_0.2_Sm_0.2_)CoO_3_ at 250 °C_._ The concept of lowering the crystallization temperature by shifting the enthalpy may, in the near future, attract more attention to the study of high-entropy catalysts.

CO catalysis plays a key role not only in the current context of oxidation in catalytic converters of exhaust gases in combustion engine, for which demand is expected to decline in the near future, but also in various other technological applications, such as high-voltage electric transmissions [44]. Recent advances in medium- and high-voltage power transmission application, particularly the shift to SF_6_-free gas-insulated systems using CO_2_ gas mixtures, underscore the importance of CO catalysis [45,46]. During combustion processes in such systems, high energy levels can induce the decomposition of CO_2_ in plasma, resulting in the formation of a variety of toxic gases, but mainly CO. This poses a serious safety risk, especially for gas-insulated system maintenance activities [47]. Traditional desiccants commonly used to adsorb moisture and other toxic substances such as HF, often prove ineffective in neutralizing CO [48]. Consequently, there is a need to develop new materials capable of catalyzing CO under these conditions. Our experiment was designed to demonstrate the potential of RECO as an effective catalytic material, especially at relatively low temperatures around 50 °C. It should be noted that our experiment was not designed to provide a comprehensive analysis of the catalytic behavior of the synthesized material; rather, it was intended to robustly assess the general feasibility of CO oxidation under mild conditions. This preliminary study lays the groundwork for future research, focusing on a more in-depth analysis of RECO catalytic properties and its potential applications, particularly in scenarios where efficient neutralization of emitted CO gas is essential, such as power transmission and other industrial conditions. 

In terms of benchmarking of the catalytic efficiency of (Y_0.2_La_0.2_Nd_0.2_Gd_0.2_Sm_0.2_)CoO_3_, previous studies have focused on LaCoO_3_ perovskite with a temperature range from 100 to 400 °C, considered the optimal operating conditions for these catalysts [49,50]. Enhanced catalytic performance shifting the operating temperature range of CO oxidation toward lower temperatures is demonstrated by Sr-doped LaCoO_3_ perovskites [51]. A recent study showed a conversion rate of 88% at 96 °C [50], whereas here, we demonstrated that RECO GT_250°C_ exhibited a conversion rate of 78% at 50 °C and 97% at 100 °C. These findings highlight the potential for further exploration of the catalytic performance of rare earth cobaltites, in particular (Y_0.2_La_0.2_Nd_0.2_Gd_0.2_Sm_0.2_)CoO_3_, which shows slightly better catalytic activity at lower temperatures compared to LCO and LSCO perovskites, while exhibiting a comparable performance to catalysts from the Cu-Ce-O system [52].

Recent mechanistic studies have shown that CO molecules exhibit optimal interaction with surface cations, particularly Co^3+^, which serve as preferred sites for CO adsorption. This finding has been confirmed both theoretically [53] and experimentally [54]. CO adsorption is followed by an oxidation process accompanied by the extraction of surface oxygen atoms, which can coordinate with the three Co^3+^ cations [55]. In light of this, the significant catalytic activity observed in high-entropy perovskite oxides underscores the importance of rare-earth metals in the 3+ oxidation state as a prerequisite for effective catalysts in CO oxidation reactions [33]. Transition metal cations, such as Co, play a direct role in providing active sites for CO adsorption and activation [56]. On the other hand, other elements in the perovskite structure, particularly the large rare-earth cations, act as “influencing agents”, tuning adsorption energies or as inert “spectators”, primarily increasing entropic contributions. Incorporating rare-earth elements into high-entropy compositions introduces a unique dimension to catalytic behavior, fostering a synergistic effect that can significantly improve catalytic performance [57,58]. However, further research is needed to comprehensively understand and describe this phenomenon beyond qualitative terms. Nevertheless, it is clear that the high modularity of high-entropy oxides requires additional research to establish guidelines for catalyst optimization involving both cations and anions.

## 5. Conclusions

In this paper, the successful synthesis of a multicomponent rare-earth cobalt oxide perovskite, (Y_0.2_La_0.2_Nd_0.2_Gd_0.2_Sm_0.2_)CoO_3_, is presented. The structural and catalytic properties of the obtained perovskite-type system were studied experimentally, which led to the following main conclusions:
The glycothermal method, carried out at a reduced temperature of only 250 °C, means a significant reduction in energy consumption of approximately 52% compared to typical high-temperature synthesis methods above 1000 °C. This low-temperature approach to synthesis demonstrates energy efficiency and sustainability, adapting to the growing demand for environmentally friendly synthesis methods. The employment of 1,4-butanediol and diethylene glycol in a glycothermal process shows significant potential for the fabrication of nanofibers from (Y_0.2_La_0.2_Nd_0.2_Gd_0.2_Sm_0.2_)CoO_3_ material. This nanostructuring method holds promise for the efficient and facile synthesis of high-entropy nanomaterials and composites.Catalytic evaluation highlights the effectiveness of (Y_0.2_La_0.2_Nd_0.2_Gd_0.2_Sm_0.2_)CoO_3_ in oxidizing CO under mild conditions, showing conversion rates of 78% and 97% at 50 °C and 100 °C, respectively, demonstrating its versatility in various catalytic processes. The inclusion of rare-earth elements in the high-entropy composition improves its catalytic properties, suggesting a synergistic effect that can significantly improve performance.


Future research should focus on optimizing synthesis parameters to further enhance catalytic performance and explore the mechanistic aspects of CO oxidation in high-entropy oxide catalysts. In addition, comprehensive studies are warranted to clarify the synergistic effects of incorporating rare-earth elements into high-entropy compositions, paving the way for the rational design and development of advanced catalytic materials.

## Figures and Tables

**Figure 1 materials-17-01883-f001:**
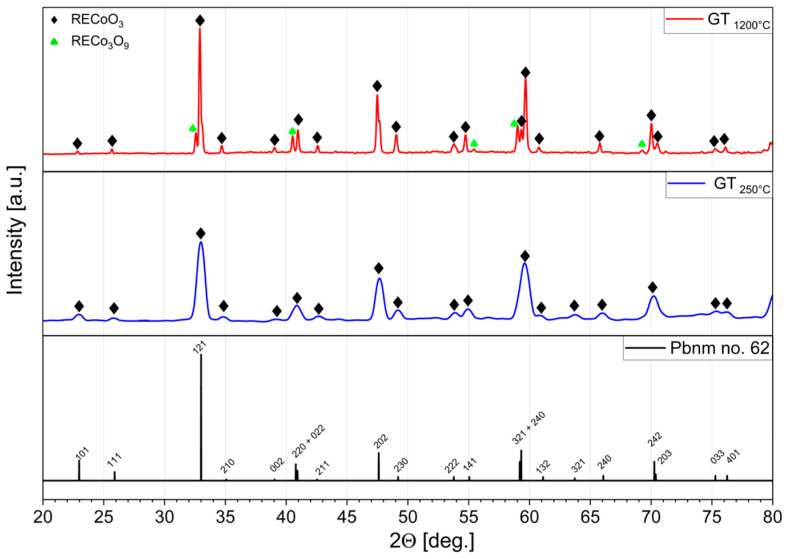
XRD patterns for powders obtained by from glycothermal synthesis without further heat treatment and cleaned by mechanical methods (RECO GT_250°C_, **—**) and the same material heat-treated at 1200 °C (RECO GT_1200°C_, **—**) with the reference model XRD pattern of Pbnm space group (Pbnm no. 62 **—**) [41].

**Figure 2 materials-17-01883-f002:**
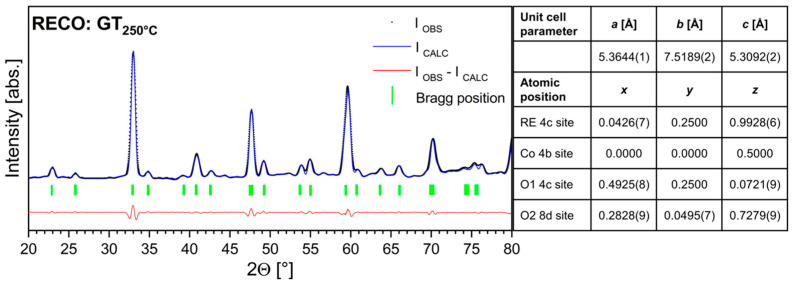
Rietveld refinement of XRD pattern of RECO sample (RE = Gd_0.2_Nd_0.2_La_0.2_Sm_0.2_Y_0.2_); (C=Co, O=O_3_) synthesized by glycothermal method at 250 °C without further thermal treatment (GT_250°C_). The Bragg position line indicates the reflections of the RECO phase (Pbnm). The table shows lattice and position parameters for RECO GT_250°C_ sample crystallized in the Pbnm orthorhombic structure (No. 62), obtained from XRD (standard deviations are given in parentheses).

**Figure 3 materials-17-01883-f003:**
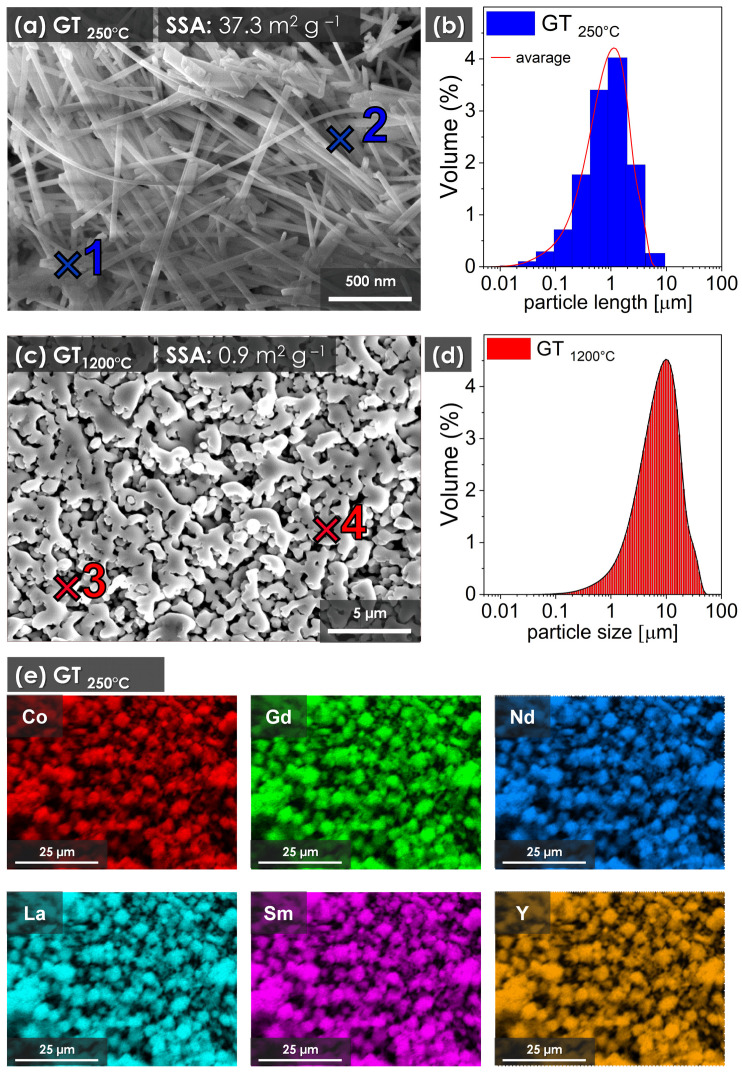
(**a**) SEM image obtained from glycothermal synthesis without further heat treatment and cleaned by mechanical methods (RECO GT_250°C_); (**b**) particle length distribution of nanofibers calculated from SEM images; (**c**) SEM image of RECO GT_1200°C_ material heat-treated at 1200 °C (RECO GT_1200°C_); (**d**) particle size distribution of RECO GT_1200°C_ sample obtained by DLS method (**e**) EDX mapping results of elements present in GT_250°C_ sample (Co, Gd, Nd, La, Sm and Y).

**Figure 4 materials-17-01883-f004:**
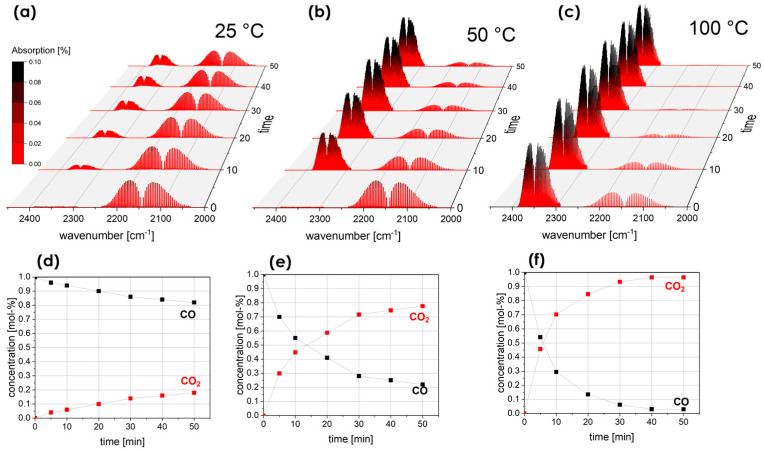
First row: FTIR spectra obtained for the gas phase inside the reactor for a reaction time of 0–50 min over the RECO GT_300°C_ material sample at different temperatures of (**a**) 25 °C; (**b**) 50 °C; and (**c**) 100 °C. Second row: CO and CO_2_ concentrations from qualitative analysis obtained from FTIR spectra as a function of reaction time for temperature (**d**) 25 °C; (**e**) 50 °C; and (**f**) 100 °C.

**Table 1 materials-17-01883-t001:** EDX analysis results for RECO material synthesized via glycothermal method without further heat treatment (GT_250°C_) and high-temperature treatment at 1200 °C (GT_1200°C_).

Sample		Co	Y	La	Nd	Gd	Sm
		at-%					
RECO GT_250°C_	1	51.1	9.6	9.4	9.8	10.0	10.1
	2	50.7	10.6	9.6	9.6	9.9	9.7
	** x^-^ **	50.8 ± 1.0	9.8 ± 0.4	9.5 ± 0.5	10.2 ± 0.5	10.3 ± 0.5	9.4 ± 0.4
RECO GT_1200°C_	3	49.7	10.1	10.3	9.7	10.6	9.6
	4	50.5	9.1	10.1	10.8	10.6	8.9
	** x^-^ **	50.3 ± 1.6	9.9 ± 0.6	9.4 ± 0.8	10.3 ± 0.9	10.2 ± 0.7	9.9 ± 0.8

Values 1–4 are derived from the data points indicated in Figure 3a,c, while the average values (**x^-^**) are calculated from a dataset comprising 20 points for each sample.

## Data Availability

The data that support the findings of this study are available from the corresponding authors upon reasonable request.

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
