# Peer review of "Synthesis and Catalytic Performance of High-Entropy Rare-Earth Perovskite Nanofibers: (Y0.2La0.2Nd0.2Gd0.2Sm0.2)CoO3 in Low-Temperature Carbon Monoxide Oxidation"

_materials, 2024, doi:10.3390/ma17081883_

Round 1

Reviewer 1 Report

Comments and Suggestions for Authors

In this manuscript, the authors reported a novel glycothermal synthesis approach that could achieve the synthesis of pure-phase high-entropy perovskite oxide at a reduced temperature of 250 C without additional high-temperature treatment. The high-entropy perovskite oxide was thoroughly studied for physicochemical properties as well as catalytic performance toward the carbon monoxide oxidation reaction. The findings can have strong implications for the rational design and development of advanced catalytic materials based on high-entropy oxides. Overall, this work has very good novelty, and the results were well presented. I would like to recommend publication pending revisions that address the below comments.

1. The last paragraph of the Introduction needs some serious revision. (1) The authors claimed that the study “aims to investigate the efficiency, performance and reproducibility of solvothermal synthesis methods”, however, the current manuscript actually focuses on “glycothermal synthesis”. (2) A few more sentences are needed to briefly highlight the main findings of this work and the implications for future research.

2. Related works on perovskite-type structures can be referenced in the Introduction (line 34-39) (e.g., Mater. Horiz., 2020, 7, 2519; Small Methods, 2018, 2, 1800071).

3. Figure 2, Rietveld refinement of XRD data analysis, the authors only showed the IOBS and ICALC, the difference (IOBS - ICALC) should also be presented.

4. Some experimental detail regarding how the Rietveld refinement was performed (e.g., what software was used, etc.) should be provided in the manuscript.

5. Recent works on high entropy oxides could be referenced in the Introduction (e.g., Energy Technology, 2022, 10, 2200573).

6. Figure 3, I assume that Figure 3a and 3e characterized the same sample, then why would the Figure 3a showed nanofiber morphology while Figure 3e did not show this type of morphology?

Author Response

Response in attached file

Reviewer 2 Report

Comments and Suggestions for Authors

The authors present a new and simple method for the preparation of high-entropy lanthanide-cobalt perovskites. The method seems to be general as demonstrated by a composition covering a large-size of lanthanide diameters, and the procedure is low-energy consuming. The prepared material is well characterised by SEM, EDX, powder diffraction and DLS, and its ability to catalyse carbon mono-oxide oxidation under mild conditions is shown. Mixed high-entropy oxidic materials are relatively modern phases with a range of potential applications, and the manuscript brings a nice alternative for their production. Therefore, I support the publication of this article. However, there is a mistake which should be corrected before the publication.

Major point: in the Abstract, a conversion of CO to CO2 99.78 and 99.97% is declared (line 18). However, it should be 78 and 3%, as the CO content in the starting mixture was 1% and the final concentration was 0.22 and 0.03%, respectively (line 231, Figure 4).

Some minor notices:

Figure 2: data points are too big and the fitting line is too thin to show a fit quality; please, change their size, or add an error line (differences in individual data points).

line180: range of lengths should be estimated, 0.62+-xx

Author Response

Response in attached file

Reviewer 3 Report

Comments and Suggestions for Authors

In the manuscript entitled "Synthesis and Catalytic Performance of High-Entropy Rare-earth Perovskite nanofibers: (Y0.2La0.2Nd0.2Gd0.2Sm0.2)CoO₃ in Low-Temperature Carbon Monoxide Oxidation"  the authors focus their investigations on two aspects.. The first one is the possibility to synthesize high-entropy perovskites by using a glycothermal method, with a notable cut in the energy costs. The second one is the oxidation of carbon monoxide by the perovskite fibers produced and characterised in the manuscript.

The materials have been characterized and references are sufficient and consistent with the discussion made in the work. Anyway, some questions need an answer before the manuscrupt could be accepted for the publication. Major revisions are required.

--------------------------------------------

Why specific activities have not been estimated? Heating and cooling experiments could give interesting info (e.g a less or more pronounced hysteresis).

What about catalytic performances of other cobalt-based perovskites in the carbon monoxide oxidation?

Given the emphasis present in the manuscript on the energy saving, have you made some estimation  of this saving?

The first paragraph needs a complete reformulation. After the first two lines, an explanatory period should follow; instead, there is a sequence of sentences not strictly correlated (and in some case difficult to understand).  Moreover: what’s the meaning of “this field of materials?”; “high entropy oxides (HEO) exploration has witnessed recent expansion”: it doesn’t sound very well; “that make them highly attractive in a variety of applications covering technological domains from…”: that make them the ideal candidates for applications in several fields:…; “but also environment and sustainability matter”: it has been already written in the first line. From my perspective, a minor editing of English language in the Introduction section is required.

Comments on the Quality of English Language

Minor editing of English language in the Introduction section (at least the first paragraph) is required

Author Response

Response in attached file

Round 2

Reviewer 3 Report

Comments and Suggestions for Authors

The authors have addressed all my comments and the quality of the manuscript has been largely increased. As a result, the paper can be considered for the publication on Materials

Decision: accept in the present form